# Sheep and Goat Meat Processed Products Quality: A Review

**DOI:** 10.3390/foods9070960

**Published:** 2020-07-20

**Authors:** Alfredo Teixeira, Severiano Silva, Cristina Guedes, Sandra Rodrigues

**Affiliations:** 1Mountain Research Centre (CIMO), Escola Superior Agrária/Instituto Politécnico de Bragança, Campus Sta Apolónia Apt, 1172 5301-855 Bragança, Portugal; srodrigues@ipb.pt; 2Veterinary and Animal Research Centre (CECAV) Universidade Trás-os-Montes e Alto Douro, Quinta de Prados, 5000-801 Vila Real, Portugal; ssilva@utad.pt (S.S.); cguedes@utad.pt (C.G.)

**Keywords:** sheep, goat, processed products, quality

## Abstract

Even though sheep and goat processed meat products are not as popular as pork, beef or poultry and are generally considered not as important, they have a very important role in meat consumption around the world. A concise review of the origin and type of the most important sheep and goat processed meat products produced in different countries and world regions is made. The manuscript also summarizes the most recent studies on sheep and goat processed meats on the physicochemical characterizations, sensory quality, microbiological quality and safety. Some conclusions and future trends in production, processing and commercial potentiality for sheep and goat processed meat products are discussed. Several possibilities exist to make them more diversified and appealing to the market. Processing meat from culled animals is an interesting way to value animals with low market acceptability. Some as fermented sausages, cured legs and pâtés have great commercial potential as highly acceptable consumer commodities. An interesting field of food research is the rediscovery of a new generation of goat and sheep meat products as functional foods that will respond to the constant innovation required by the meat industry. Everything related to food safety must be considered in the future.

## 1. Introduction

Despite not being the most consumed meat in the world, in the decade from 1994 to 2004, there was a notable increase of 75% and 42% in the production of sheep and goats, respectively, with a trend that continued until 2018 [1]. Countries with a long tradition of consuming sheep and goat meat also consume many products, such as hams, sausages and pates, or other processed products. Many of these products correspond to ancient methods of preserving meat at a time when there was no other way of preserving than curing with salts, air drying or smoking. Smoking, drying and salting are the oldest ways to preserve the meat, and particularly, some of them are nowadays recognized as protected origin designation (POD) or geographical protected indication (GPI) brands, as has been mentioned by Teixeira and Rodrigues [2]. Some of them are linked to the peculiar consumption traditions of some countries. For example, in Northern Europe, particularly in Scandinavian countries, there are typical and traditional dry-cured sheep and goat meat products, such as Fenalår and Pinnekjøtt in Norway or Hangikjøt and Skerpikjøt in Iceland and the Faeroe Islands [3]. In countries where the religious traditions prohibit the use of pork meat in processed Halal products, meat and fatty tissues from sheep and goats, as well as the tail fat from sheep breeds, are used, as in Turkish sucuk [4] and the Algerian, Moroccan and Tunisian Kaddid or Gueddid [5]. In the Northeast of Brazil, specifically in Petrolina-Pernambuco but, also, in Ceará, the carcasses of boneless and semi-dehydrated sheep and goats and, recently, the product called “Manta de Petrolina” have been distinguished as cultural heritage and registered as a quality brand. In some Mediterranean European countries, young lambs and kids producing light carcasses are highly appreciated by consumers, and many of them are very popular PDO (protected designation origin) or PGI (protected geographical indication) brands [6]. The meat of sheep and goats coming out of these quality marks, as well as culled animals, are cured and processed, producing popular products such as the Spanish Cecina de castron [7] or the Italian violin di capra [8]. Recently, studies on the use of goat and sheep meats in the production of new processed products and methods [9,10,11,12,13,14,15,16,17] confirm the importance of research and innovation as key factors in advances in the production of sheep and goats, particularly in the control of food processes, physicochemical characterization, food safety and the sensorial properties of new products [18]. Thus, the purpose of this review is to summarize the most recent studies on the quality of sheep and goat processed meat products and to contribute to a better awareness and spreading the information about these products.

## 2. Sheep and Goat Meat Processed Products

The need to preserve meat for later consumption dates back to ancient times. Salting, drying and smoking are the oldest forms to preserve and keep meat in the world. Fermentation and ripening together is another efficient methodology with no cooling required or other processes of meat preservation also used a long time ago to increase the shelf life.

Currently, the processing of meat is not exclusively for the need for preservation but, mostly, to satisfy consumer demand and the acceptability for products with traditional palatability and flavor characteristics. Given the huge variety of existing products industrial or handmade-manufactured, we decided to group the products into three groups: dried and semi-dried products, sausages and cooked or precooked prepared products.

### 2.1. Dried and Semi-Dried Products

All products that are naturally or artificially dehydrated, salted, cured, smoked or not will be considered as dry and semidry products. Some fermented sausages with short or longer ripening periods will also be part of this group.

In South America, dried meat strips from several animal species, including sheep and goats exposed to sun, wind and smoke, are called jerky or charqui. According to Fadda and Vignolo [19], the charqui is made from small pieces of fresh meat salted and pressed for several days, dried and with most of the water removed and that can be stored for several months without the need for refrigeration. In Northern Brazil, it is very common to have goat and sheep charqui called Carne de sol. In Northeast Brazil, ripened and cured goat or sheep meat locally known as manta correspond to a deboned carcass, only maintaining the rib bones and cutting the main muscles into thin meat layers, exposing them, resembled a blanket (manta is the Portuguese word that means blanket) that is lightly salted and hung in boxes protected with anti-fly nets and sun-dried. The most famous product is the Manta de Petrolina and the CYTED (Programa Iberoamericano de Ciencia y Tecnología para el Desarrollo) Iberian American Network of Quality Brands of Ibero American Meat and Meat Products (MARCARNE), which sponsored and promoted events in the past two years to recognize and protect its geographical designation.

One of Norway’s most important dry-cured meat products, with a long tradition of consumption since the time of the Vikings, is Fenalår, a dry-cured ram leg or mutton, and the Pinnekjøtt, a dry-cured side of lamb or mutton [3]. The designation Fenalår derives from the old Norwegian words fenad and lår used to designate mutton and leg. Legs are brine- or pickle-curing. Traditionally the salted legs were dried and hanging from pillars in storehouses and smoked, but today, production plants with control of the temperature, relative humidity and air velocity are used. Smoke today is slightly used only to add taste and flavor. The Pinnekjøtt is a dry-cured meat from the side of lamb or mutton placed in boxes in layers of meat and coarse salt for 1.5 to 3–4 days and rinsed in freshwater to prevent the salt precipitation during the dry period: 10 to 15 days in a room at 10–15 °C temperature and 65–75% of relative humidity or 6–7 weeks if pre-tending a more dry product.

Hangikjø, an Iceland-smoked shoulder or leg of lamb, is consumed in the festive meals of Christmas and Easter [20]. The legs and forequarters are salted and dried or immersed in brine for 1–6 days and smoked. Traditionally, they are always smoked with dried sheep manure and birchwood. In Iceland, there are different preferences for the various types of Hangikjöt, and the main trends are actually the use of less salt and less smoke, but dried sheep manure is always needed, because the smoke gives the meat its unique and different flavor [21]. Furthermore, in Iceland, there are still other soured (pickled) and fermented lamb meat products: Blóðmör—a blood sausage, Lifrarpylsa—a liver sausage, Lundabaggi—the rolled lamb flanks, Bringukollar—a lamb brisket, Hrútspungar—the pressed testicles and Sviðasulta—a head cheese from boiled singed sheep heads [20]. Skerpikjøt, wind-dried fermented lamb, as well the Ræstkjøt (semi-dried mutton), are two popular meat products from the Faroe Islands and are especially highly valued [22]. Carcasses are opened up and flattened, cutting through the ribs on one side close to the backbone and hung up along the rivers or brooks on low banks or places with good air circulation [3]. Additionally, being a traditional dish from Faroe Islands, Sperðil is made from the tallow around the rectum of sheep and prepared into a type of primitive sausage that can be used as spread on bread, eaten with fish or added to the traditional unleavened bread [22].

In Asia, mainly in the Himalayan area (Nepal, Northeastern India, Tibet and Bhutan), there are different fermented meat products, using pork, beef, yak and sheep and goat meats [23]. Suka ko Masu is a dried/smoked product usually consumed in Nepal and India. The dried and smoked goat and buffalo meats [24] are sliced in strips of 25–30 cm hanging above the kitchen oven for 7 to 10 days and are consumed mixed with mustard oil, turmeric powder and salt. In the Kumanu Himalayas, the greema/arija is a popular semidry sausage [25]. Some other ethnic meat products, such as kheuri in Sikkim India, are a mixture of yak or beef stuffed and pressed into the sheep stomach and hung in open-air space for one to two months [24,25].

Pastrami is a traditional Romanian dry meat product without heat cure preserved by nitrite or nitrate salts and seasoned with fenugreek and garlic normally made by beef and buffalo water and, in Egypt, also from sheep, goat and camel meats [15].

Over the last few years, in the Mediterranean area of Europe, several goat or sheep processed products have been produced. Mainly in Spain but, also, in other European countries, such as Italy, culled goats meat are, salted, smoked and air-dried following a recipe for cured ham established 2000 years ago as Cecina, after the Latin siccina, which means cured meat, known as Cecina de cabra and Cecina de castron made from goat meat legs [7] and, also, Violin di capra, an Italian traditional goat dry-cured leg [8].

The Kaddid or Gueddid is a traditional salty and dry meat product typically prepared from mutton meat salted and dried outdoors in Maghreb countries (Tunisia, Algeria and Morocco) [5]. The Gueddid can be stored at room temperature for more than a year, and, before consumption, it is desalted after immersion in water, being used in various dishes, such as Aiche, Couscous, Elhessa and Marloga in Algeria [26]. In addition to Gueddid, there are other dry and fermented or not fermented products of North Africa and the Mediterranean of sheep and goat meats, such as Kourdass (a sausage prepared from offal lamb stomach, intestines, liver, lung, spleen and fat), Tidkit (a sun-dried product), Boubnita (a dry lamb fermented sausage), Soudjouk or sucuk (a dry fermented and uncooked sausage of Turkish origin) or Maynama and Merdouma (smoked products) [5]

In Southern Africa, a goat meat drying technique using salting and smoking processes and known as chinkui in Mozambique’s Northern Tete Province is also commonly used [27] as a complementary source of protein in local rural areas of developing countries with large undernourished populations.

The large amounts of dry and semidry sheep and goat products worldwide, with a special focus on the most depressed areas of agriculture and livestock in the world, are evidence of the importance of sheep and goat productions as a way to overcome protein deficits using food preservation ways that respect the environment, producing and consuming meat products with low carbon footprints.

### 2.2. Sausages

Although not so common in sheep or goat meats, there are some sausages traditionally recognized for their individuality and linked to the consumption history in the regions that produce them. Some of them are fermented sausages fresh or smoked. In Middle Eastern countries, fermented sausages are produced from many different animals instead of pork (beef, buffalo, camel, horse, lamb, goat and mutton) [28,29,30]. Sucuk, also known as sujuk, is a dry-fermented Turkish sausage very common in Southeastern Europe, the Middle East and Middle Asia [30]. Sucuk and other similar sausages are made of beef, goat and sheep meats and tail fat [31], with garlic, salt and other ingredients such as sugar, nitrites and/or nitrates and various seasonings and spices [32]. Sheep tail fat is a constituent of Turkish fermented sausages and kebabs.

Fjellmorr and Lambaspaeipylsa, are dried fermented sausages traditionally produced in Norway and Iceland that contain lamb, as well as beef and pork [33]. Another Norwegian product is Fårepølse, a mutton sausage made with a large meat variety, including goat [34].

Arija and Geema (jamma) are traditional goat meat products from Kumanu Himalaya in Northern India. To prepare Geema, the meat goat is minced and mixed with salt, wild pepper, chili powder, water and fresh blood. This mixture is stuffed into small goat intestines, and then, the sausages are boiled and then are dried for a period of 15–20 days hung above the kitchen oven. The Arija is the same process, but the goat meat is mixed with goat lungs, and the mixture is stuffed into goat large intestines [22,35].

Several studies are investigating the incorporation of sheep or goat meat in sausages. In Vienna sausages, a cured and smoked product [36], a study was conducted to evaluate the consumer acceptability of goat meat frankfurters using three different diets, including canola oil or beef fat [37], using different levels of pork fat in goat mortadella [38] and using sheep tail fat and lean mutton for manufactured dried fermented sausages and studying the effects of the combined starter and species in physicochemical and microbiological properties [39] and studying the suitability of goat meat to the restructuring techniques, assessing the quality during refrigerated storage of the cured goat product [40,41]. In a study, meats of sheep and goats from culled animals were pointed out as valuable alternatives to that meat-producing sausages with different levels of pork fat [9]. The nutritional characteristics and consumer acceptability of different combinations of goat meat with beef have been studied [42], and the technological properties in frankfurter productions were studied using meat from culled animals [13].

Rediscovering processed products based on sheep and goat meats, incorporating functional additives and improving their nutritional and food quality, is increasingly a practice that may add value to meats of low commercial values.

### 2.3. Cooked or Precooked Prepared Products

The process for producing sliced, easy-to-prepare, prepackaged food products from a goat or a sheep meat is not common. However, we found some examples of processed sheep and goat meat products that can be considered cooked or precooked products. In Northeast Brazil, there is a product called “Buchada de Bode” (buchada coming from the Portuguese word bucho, the animal’s stomach, and bode is the Portuguese designation of the male goat) made with reticulum stuffed with entrails or organs, such as blood, intestines, liver and lungs, seasoned and then sewn up and cooked.

In Turkey, the traditional meat used to make kebabs is lamb. There are several types of kebabs, but the world-famous one is the Döner, made conventionally from lamb, which is slowly roasted on a vertical spit and then thinly sliced off. In local markets, it is very popular to order precooked kebabs of minced lamb mixed with tail fat and vegetables such as eggplants, tomatoes, peppers and onions, ready to roast at home. In India and Pakistan, there are many varieties of kebabs commonly prepared with lamb. In reality, the consumption of kebabs has now extended to the whole world but occupies a prominent place in the Mediterranean area.

Lamb and goat are the most popular meats used in Arab countries, where there are some popular street-prepared dishes such as Shawarma (a food made with marinated cuts of meat skewered on a vertical rotary roasting fork (spit) and baked by radiant heat), Kibbeh or kebbah (a traditional food consisting of a mixture of ground bulgur and minced lamb or beef meat, which are made as a dough) or the Kofta (another dish made with minced meat) [43].

In the Mediterranean countries of North Africa, there are several cooked and crystallized products from sheep and goats seasoned and stuffed with local species, such as Laknaf, tha Cachir and Khliaa ezir in Algeria; Osbana in Algeria Tunisia, Libya and Morocco; kabiba in Egypt; Mcharmia in Algeria and Morocco; Boubnit or Membar in Algeria and Egypt; Mkila, Tehal and Tangia in Morocco; Bekbouka in Morocco, Algeria and Tunisia or Mrouzia in Morocco and Tunisia [5].

In Mexico, barbacoa is a very traditional dish prepared with lamb or goat, which is slow-cooked in an underground oven lined with hot coals and covered with agave leaves until very tender and juicy. Today there are some food industries that commercialize precooked barbacoa ready to eat at home and that can be exported mainly to countries with Mexicans immigrants.

In Spain, one of the most famous dishes is “Lechazo de castilla”, a milk-fed lamb slow-roasted in firewood ovens. Corresponding to modern lifestyles, there are also food industries that offer a delicatessen product, such as Lechazo, in a precooked and vacuum-packed form, ready to serve on the table in 30 min.

Chanfana is an old, traditional Portuguese dish cooked in black clay pots made from local clay inside a wood oven slowly cooking old goat or sheep meats in red wine with some herbs and spices, making the meat tender and more palatable. The resulting dish turned out to be so tasty that it is now a regional delicacy in Portugal.

In the last years, several studies using sheep and goat meats with cooked or precooked products and some products ready to cook have been developed as patties [44,45,46,47], nuggets [48,49,50,51,52,53,54], hamburgers [55,56] or pâté [11,57,58,59] and others [60,61,62,63]. In the Indian subcontinent (India, Pakistan, Bangladesh, Nepal and Sri Lanka), there are several traditional meat products and recipes using sheep and goat meats, such as Rogan Josh, a lamb meat marinated with Kashmiri chilies, Aab Gosht, a lamb curry cooked in milk, Rista and Goshtaba, a minced lamb meat cooked in mutton stock gravy, Haleem prepared from goat or buffalo meats or Kolhapuri mutton with different curries [64].

Unfortunately, many of these products are unknown to the general public, but they are the basis of foods for populations in developing countries, with peculiar sensory qualities and, at the same time, with high nutritional values and nutraceutical characteristics. Most of them deserve to be recognized as cultural heritages of the gastronomy of these countries and with world recognition with support and protection.

## 3. Meat Processed Product Qualities

The set of properties normally used to characterize the quality of a meat product are associated with physical and chemical attributes, with our sensory feelings and related with the perception that it is safe to eat. The quality of the physicochemical, microbiological and safety and sensory properties of sheep and goat processed products will be covered in the following sections.

### 3.1. Physicochemical Quality

The physicochemical characteristics of many of the products described above have been studied in the last 10 to 15 years. Table 1 summarizes the main attributes measured and effects studied on different processed goat and sheep products. Most of these products show high protein contents, balanced fat and fatty acid profiles, low cholesterol contents and resistances to lipid oxidation. The color parameters in the different meat-processing technologies show some pigment oxidations as a result of the preservation methods. The color parameters H* and C* were affected by salting and air drying, and the meat-based products became darker. Globally, the process methods improve the preservation quality once the water activity (a_w_) is reduced.

Some of the studies in some of these new processed products showed interesting results to improve the physicochemical quality. Some studies pointed out the importance of using meats from culled sheep and goats to obtain processed products, with added values to animals with very low commercial values and acceptability by consumers [9,10,11,12,13].

One of the disadvantages of processed meat products is the oxidation of the lipid fraction and proteins. Several studies have been carried out to reduce this effect. Using fermented goat sausages, it was observed that preparations containing 0.05% rosemary showed better oxidative stability with lower values of thiobarbituric acid reactive substances (TBARS) than those with only 0.025% rosemary [74]. In cooked meat patties, the antioxidant potentials of kinnow rind, pomegranate rind and seed powders extracts were evaluated [44]. In addition, the antioxidant potential of broccoli extract in goat meat nuggets was evaluated and validated [47]. In cured restructured goat meat product, it was observed that some natural antioxidants have relevant antioxidant activities. For example, sodium ascorbate and alphatocopherol acetate improved the color and lipid stability by reducing the lipid oxidation and free fatty acids values [39]. Additionally, the use of the antioxidant activity of hop powder and a hop infusion in lamb patties was referred [45]; the inclusion of oregano extract in the manufacture of cooked sausages showed lower lipid and protein autoxidation [73], as well as the attributed antioxidant activities of chrysanthemum morifolium flower to goat meat patties [46].

Besides the more traditional products linked to the culture of the people and the ancestor methods of preserving meat, in general, the results indicate that the meats of sheep and goats show aptitudes to processing. The sheep and goat meats can be processed in cured, dry and fermented products or to incorporate meat mixtures in patties, pâtés, sausages and hamburgers as a way to add value to meats from animals that do not meet the quality specifications of products, such as PDO and PGI brands.

### 3.2. Microbiological Quality and Safety

Food safety is one of the main concerns to the food industry, government food safety regulatory authorities and consumers due to the significant increase in foodborne diseases and outbreaks reported worldwide in the 20th and 21st centuries. Microbiological, chemical and physical risks are the main group of risks to food safety in the food industry. The microbiological hazards involve foodborne pathogens; chemical hazards are related to antibiotic, pesticide and herbicide issues and physical hazards are associated with strange objects in foods that can cause injury or illness to consumers. This subsection will focus on microbiological topics related to food safety.

Related to food safety, fermentation plays the following roles in food processing [75]: (1) food preservation by inhibitory metabolites, such as organic acid; formation (lactic acid, acetic acid, formic acid and propionic acid); ethanol; bacteriocins, etc., often combined with a a_w_ decrease (by ripening or salting) [76,77]; (2) food safety improvement through pathogen inhibitions [78,79] or toxic compound removals [80]. Recent studies on fermented meat products confirmed the existence of a great fount of microorganisms with probiotic characteristics in those products [81], guaranteeing hygienic and safety quality [82]. Studying ripening and fermentation effects on the meat microflora of diverse species (cattle, horse, wild boar, goat and deer), an inhibitory effect of a_w_, pH and lactic acid bacteria (LAB) was found on the pathogenic bacteria during the fermentation process [83]. In the raw materials, the same authors working with uncooked meat products observed the presence of typical flora such as Staphylococcus aureus and coliforms in all samples, and none had *Salmonella* or *L. monocytogenes*. However, at the end of the fermentation process, an increase in LAB was observed. This increase has an opposed action on the contaminating flora.

Salting has been used as a food preservative for thousands of years. The pH and a_w_ numbers revealed that processing could have an essential role to control the spoiling of meat, promoting the safety and shelf-life stability of the products exposed to microbiological growths. For Norwegian Fenalår, 5–10% NaCl was registered [84]. Skerpikjøt with a a_w_ value 0.90 had no salt addition [84], which was directly related to botulism incidences [85]. The final content of 4.5% NaCl (or lower) in sheep ham was challenging, considering unwanted bacterial growths, but preferable, considering health recommendations [85]. The salt content depends on production practices; for Spanish lamb ham, it was referenced 7.96% NaCl and a a_w_ of 0.88 [45], for dry-cured goat legs, values of 3.8% NaCl and a a_w_ of 0.83 and 4.7% and 0.87 for sheep after salting and air-drying processes were found [10]. Nitrite salt was used in pâté production [59], and processed goat hams were dry-salted with nitrite salt (sodium chloride and sodium nitrite) [65]. Nitrites may be associated with cancer and other health issues [86]. However, the nitrite contents in the pâtés and cured hams were lower than the maximum recommended by the FAO [87], and the European Union allowed their inclusion as food additives [88]. Microbiological safety implications in salt-reduced Fenalår was shown to be important [89], especially when no nitrite was added, because of the considerable increase of the growth potential of *L. monocytogenes.*

A study [40] on the effects of natural antioxidants on the quality of cured, restructured goat meat products during refrigeration storage pointed out the effects of sodium ascorbate and alpha tocopherol acetate on microbiological properties. No significant effects were found in the microbial counts, and coliforms and *Staphylococcus aureus* were found occasionally during the refrigerated storage period. Microbial counts were reduced by the addition of extracts obtained from grape and olive pomaces [90], showing the possibility of using them as a sodium ascorbate replacer in lamb meat products.

The effects of avishane shirazi (*Zataria multiflora*) and clove (*Syzygium aromaticum*) essential oils on microbiological properties of ground sheep meat during refrigerated storage were evaluated [91]. Total mesophilic counts (TMC), LAB, coliforms, yeasts and *Listeria* counts were performed. The most significant effect of avshane essential oil (AEO) was observed in LAB growth-controlling (1.2–3.4 log cycles) and coliforms (1–2.5 log cycles). The inhibitory effects of oils on the TMC and yeast counts were minor, while both oils had strong and comparable preventing effects on the Listeria populations.

Mycobiota and mycotoxins in Portuguese pork, goat and sheep dry-cured hams were studied [12]. The contamination of Portuguese-ripened meat products with ochratoxin A (OTA) and aflatoxin B1 (AFB1) was examined, and the contamination potential cause that fungi established. No OTA was detected in the goat and sheep samples. So, OTA contamination does not appear to be a hazard in small meat joints during short ripening periods in products such as goat or sheep legs.

### 3.3. Sensory Quality

The nutritional and physicochemical characteristics play an essential role in food production. However, their organoleptic characteristics and acceptance by consumers will assure their marketplace. The sensory analysis allowed us to define and predict food products’ acceptances by consumers. A review about sheep and goat-processed products was conducted [18], referring to several studies on the sensory quality of new sheep and goat products. Some different treatments or processes were tested and compared. Differences between sheep and goat products [10,92,93,94] were found. Globally, sheep products were juicier, and goat products were harder, had more intense flavor and more acid and rancid tastes. More curing time led to higher brightness. Variations in the processing methods in distinct locations produced differences in sheep ham sensory characteristics [85]. The effect of tumbling after dry-salting and the processing time on the sensory characteristics of dry-cured lamb legs [45] was not significant between no, short and long tumbling, except for pastiness, which should not be considered an eating quality defect, since it was not that significant. Storage time and packaging can influence the quality of lamb pâté prepared with “variety meat” [59], and a decrease in sensory texture and overall impression could reduce the shelf life.

Table 2 summarizes the primary sensory studies performed by trained panelists on new meat processed products incorporating sheep and goat meats.

Artificial food additives (e.g., antioxidants) used to modify sensory qualities can have health implications [97,98,99,100,101,102] concerning both consumers and food manufacturers. Consequently, a revived appeal about natural products and its research has grown [103,104,105]. The effects on sensory aspects were investigated. Some of the impacts were color and odor preservation, the inhibition of discoloration and off-odor formation; a decrease in oxidation odor and flavor and changes in the overall acceptability. It has been shown [96] that hops could be applied to make lean lamb patties to minimize the deterioration of flavor by oxidation. In addition, a higher flavor intensity was observed when hop was used as a powder compared to an infusion or was not used. The use of three distinct concentrations of *Origanum vulgare* extract, with potential antioxidant properties, in sheep sausages [73] showed no significant differences in smell and taste between the various levels of oregano and sodium erythorbate. However, sodium erythrobate and higher levels of oregano got a more intense red color than the control, possibly decreasing the acceptability by consumers.

The use of paprika influenced the sensory characteristics as a persistence and intensity of flavor, spiciness and off-odor in ewe and goat meat sausages [92]. Exploring the possibility of using goat meat in traditional Sucuk production, no significant differences in cut appearance, color and odor between goat and beef Sucuk were observed [95]. To soothe the goat flavor, making a more acceptable product to consumers, the authors suggested replacing goat fat with beef fat. The sensory quality of culled goat meat frankfurters [13] showed very close grades between the goat and beef (CON) frankfurters, and no significant differences were noticed for odor, taste, hardness and juiciness. Color was the single attribute significantly different in frankfurter formulations. Goat frankfurters were lighter than beef; the more goat content, the lights were the sausages.

Taste panels were used to objectively evaluate the food. Human beings act as machines, but they are not machines, and sensory evaluations always have some subjective issues. Much more variability is expected to be found in sensory than physical-chemical analyses; however, with the use of standards and references, and with the appropriate statistical analysis, it is possible to reduce the variability and have a reliable sensory characterization of the food.

Nevertheless, sensory characteristics are not the only essential point of a food product, and consumers have the final word. So, consumer panels play an important role informing about their likes, dislikes and preferences. Following, a set of studies are presented informing about acceptability and preferences considering sheep and goat meat products.

A sensory acceptance and purchase intentions by 80 potential consumers of goat mortadella with distinct fat and goat meat percentages study [38] revealed the possibility of making an agreeable value-added goat meat product. Consumers preferred the goat mortadella with less fat and more goat meat in all the studied sensory aspects, except texture.

Sheep meat sausages were referred to as an unexploited market opportunity when the sensory characteristics of fermented, cured sausages manufactured with beef, pork and sheep meat were evaluated [106]. The reason was that there were no commercial examples, and no significant differences in texture (hardness, springiness, adhesiveness and cohesiveness) after anaerobic fermentation were observed among the species. Additionally, just small differences were noticed in the colors. Nevertheless, even if not tested, it is alleged that consumers would be capable of finding a texture difference due to distinct fat-melting points, the highest for sheep meat. Additionally, a sensory analysis was completed to identify the possibility of the coating or even removing the peculiar sheep meat flavor to please consumers unfamiliar with this product [106]. An extreme characteristic was simulated by producing a sheep meat and beef-blended sausage: spicing it or not. The authors also variably added sodium nitrite and a garlic/rosemary flavor. A spiked sheep meat flavor produced a general significant decline in mean liking. Still, an increase was detected when garlic/rosemary was included. Nitrite did not affect the liking. Conclusions hinted at the option of hiding sheep meat flavor to attract unused consumers. Commercial exemplars might be prepared for these consumers, but the required “mutton” name would adversely affect expectations in some markets.

Table 3 summarizes the sensory studies with nontrained consumers on brand new manufactured products using sheep and goat meats.

The improvement in the uniformity of flavor, appearance and texture of fermented products by the appropriate physiologically active starter culture was already mentioned in 1970 [110]. A sensory analysis of fermented mutton sausage using different starter cultures [111] exhibited satisfactory records in refrigerated 60 days of storage. Additionally, distinct starter cultures [112] generated median values in a nine-point scale for goat meat-fermented sausages’ global acceptability, and no significant differences for all treatments were observed. Judges used some sporadic remarks such as “rancid” and “soap” possibly linked to the product’s fat oxidation or even to the lipolytic action of starter culture microorganisms. Lactic starter cultures were used to study the effects of fermentation and drying temperatures on the sensory attributes of goat meat dry sausages [113]. Fermented sausages at a temperature of 30 °C, and dried at 10 °C, presented higher satisfactoriness concerning taste, flavor, texture and overall acceptability in a five-point hedonic scale.

Consumers generally appreciated fresh sausages made using sheep and goat meats [92], showing no evident predilections for sheep, goat or seasoning. The hedonic test completed to investigate the use of hop as an antioxidant in lean lamb patties [96] showed that the control and infused patty flavors had significantly higher scores than powdered patty flavors. Several elements of hop essential oil and resin are the cause of its strong odor and bitter taste [114,115]. A remarkable effect of two essential oils on the color and odor attributes of ground sheep meat [91] was registered. The overall acceptability was not affected.

Examining distinct fat content additions, which can be challenging, on the sensory acceptability of a goat meat-fermented sausage [112], the assessed characteristics were not significantly different. Additionally, no significant differences were found in consumers’ preferences concerning meat [58] substitutions (pork by sheep) on cooked ham-type pâté or fat [94]. However, significant differences were mentioned [9] in the overall acceptability of sheep and goat sausages with distinct fat amounts. Consumers preferred higher fat content goat sausages. Dry-cured sheep meat acceptability by native and immigrant consumers in Spain was investigated in a cross-cultural study [108]. Independently from their cultural background, consumers appreciated “Cecina” from culled ewes. Generally, the overall acceptability was not modified by the animals’ finishing levels, even though leaner animals were preferred by some consumer clusters.

A positive consumer attitude towards salt reduction and the use of natural antioxidants [109] emphasizes the need to develop improved meat-processing techniques created from scientific information with potential benefits for the meat industry and public health. Additionally, the knowledge about consumers’ perceptions for healthier products requires the inclusion of consumption contexts. The way a product is consumed is important, as it can lead to different results [109].

Four culled goat meat percentages (G25, G50, G75 and G100) were used to produce frankfurters and compared to beef (CON) [13]. Consumers observed an increasing lightness and decreased redness in frankfurters with increasing goat meat contents, but it was not negatively perceived. More than 80% of consumers identified goat frankfurters and the CON with a pleasant color in a Check-all-that-apply (CATA) analysis. Similar results were pointed out for pleasant appearance, pleasant odor, tasty, soft and juicy. Additionally, no atypical taste and odor were marked. Panelists preferred G50, while consumers preferred G75.

Globally, sheep and goat meat products reported in the previous studies were well-accepted and proven to be a way of giving added value to less-appreciated products. The use of additives can mask some unpleasant characteristics, such as odor, for example. However, there are always consumers that appreciate that peculiar odor, and the complete removal of the innate sheep and goat characteristics can be a negative point in food processing, because they lose their identity.

## 4. Conclusions

Several sheep and goat processed meat products confirm the importance of these species in consumer cultures around the world, and most of them have a great demand and are very well-appreciated in many countries and regions, such as North Europe, the Mediterranean area, Middle East, North Africa or Central Asia. In countries without a tradition of goat or sheep consumption, they are frequent in delicatessen stores or ethnic markets. Processed products such as sausages, pâtés or frankfurters, blending different animal or vegetal fat sources, can give additional value to the less-appreciated goat meat from culled animals. Using some natural antioxidant byproducts in meat processing, sheep and goat products can be used as functional health-promoting foods and can also improve the shelf life, product color and reduce the lipid and protein autoxidation. The rediscovery of a new generation of goat and sheep meat products as functional foods and eating quality is an exciting food research field, answering to the constant innovation requirements by the meat industry. Several possibilities exist to process sheep and goat meats to make them more diversified and appealing to the market.

The use of starter cultures, spices, essential oils and other additives may provide nutritional and sensory advantages of dry fermented sheep and goat sausages. Cured and smoked goat and sheep legs have great commercial potential, with a highly acceptable consumer commodity.

Traditional goat and sheep meat products are part of the cultures of their countries, and many of them must be better studied and characterized as a way to preserve and protect them in terms of certifications of their origins. The deep knowledge of the formulations, the optimization of processing methods, improving advancing packaging and preservation procedures and the organization of the distribution and marketing chains are actions that should be prioritized. Everything related to food safety and the increasing importance of traceability is detailed information to consumers and will continue to be a matter of constant concern and should be considered in the future.

## Figures and Tables

**Table 1 foods-09-00960-t001:** Summary of the principal studies and attributes measured on the physicochemical characterizations of processed sheep and goat meat products.

Product	Attributes Measured	Effects Studied	References
Restructured goat meat	Color, pH, moisture, protein, fat, ash, TBARS	Natural antioxidants on quality	[40]
Lamb burgers	Moisture, protein, fat, ash, fatty acid profile	Chemical and sensory quality	[55]
Lamb hamburger	Moisture, protein, fat, ash, TBARS, cholesterol, pH, a_W_, cooking and texture characteristics	Cotton level on chemical and sensory quality	[56]
Goat nuggets	pH, color, moisture, protein, fat, ash, texture, TBARS	Antioxidant of bael pulp	[49]
Goat pâté	Moisture, protein, fat, ash, color, pH, a_W_, minerals	Variety meat	[57]
Lamb pâté	pH, a_W_, moisture, protein, fat, color, nitrites	Storage time on quality	[59]
Sheep and goat pâtés	pH, a_W_, dry matter, ashes, protein, fat, collagen, cholesterol, fatty acid profile	Fat sources and proportions on quality	[11]
Goat patties	pH, moisture, protein, fat, ash, texture	Incorporation of goat fat on quality	[44]
Smoked goat ham	pH, moisture, protein, fat, ash, NaCl, nitrites, fatty acid profile, volatile substances	Diet effect on nutritional value	[65]
Cooked sheep ham pâté	pH, a_W_, moisture, protein, fat, ashes, color, TBARS	% sheep meat in mixture on quality	[58]
Goat cured meat	pH, a_W_, water holding capacity, color, myoglobin, texture	Salting and ripening on meat quality	[66]
Dry-cured Halal goat meat	NaCl, pH, a_W_	Salting time on meat quality	[67]
Dry-cured lamb leg	pH, ash, moisture, protein, fat, NaCl, free fatty acids, volatile substances	Tumbling on meat quality	[46]
Goat “mantas”	pH, a_W_, moisture, protein, fat and fatty acids, ashes, TBARS	Added value to culled animals	[68]
Goat and sheep “mantas”	pH, a_W_, water holding capacity, color, texture	Species	[69]
Goat and sheep cured legs	pH, ash, moisture, fat, protein, fatty acids, TBARS, NaCl, color, collagen, cholesterol	Species, salting and ripening time	[10]
Goat frankfurters	Moisture, protein, fat, texture	Source of fat on texture acceptability	[37]
Mutton sausages	Moisture, a_W_, pH, fat	Different starter cultures on quality	[39]
Goat Mortadella	Moisture, ashes, protein, fat, color, pH, a_W_	Pork fat level	[38]
Sheep and goat Sucuk	Moisture, fat, protein, ashes, color	Effect of sheep and goat	[30]
Goat blood sausage	Moisture, fat, protein, ashes, fatty acids, amino acids cholesterol	Descriptive	[70]
“Buchada” goat	pH, a_W_, moisture, protein, fat, cholesterol, amino acids	Descriptive	[71]
Chevon salamis	Moisture, protein, extract ether, ashes,	Descriptive	[72]
Sheep pastrami	Warner-Bratzler shear force	Vacuuming pressuring on tenderizing	[14]
Sheep and goat sausages	pH, a_W_, moisture, protein, fat and fatty acids, ashes, TBARS	Different pork back fat levels added	[9]
Goat sausages	Moisture, fat, protein, ashes, fatty acids, cholesterol	Combinations with beef	[42]
Sheep sausages	pH, color, texture, TBARS, free fatty acids, volatile compounds	Oregano concentrations	[73]
Goat frankfurter	pH, ash, moisture, protein, fat, color, texture, fatty acid profile	Level goat meat	[13]

TBARS—thiobarbituric acid reactive substances and a_w_—water activity.

**Table 2 foods-09-00960-t002:** Principal studies on the sensory evaluations of new products from sheep and goat meats with trained or experienced panelists.

Product	Species	Attributes	Panelists (*n*)	Training	Scale	Reference
Dry-fermented sausages	Goat and beef	Appearance, cut appearance, color, odor, texture and taste	9	Previous experience	1—extremely unacceptable; 9—extremely acceptable	[95]
Dry-cured leg	Lamb	Flavor and texture	12	Previous experience and two one-hour sessions	1—lowest; 5—highest	[46]
Fresh sausages with and without paprika	Sheep and goat	Odor intensity, off-odor presence, flavor, off-flavor, hardness, juiciness, fibers presence, spiciness and sweetness	9	Trained	Structured, unnumbered scale of 10 cm. Minimum (no sensation) to maximum (extremely intense sensation)	[92]
Patties cooked	Lamb	Oxidized flavor intensity	7	Trained two one-hour sessions	Ranking test. 5-point descriptive scale, from 1 to 5 denoted imperceptible or extremely high oxidized flavor, respectively	[96]
Freshly cooked patties	Lamb	Hop flavor intensity	6	Trained one half-hour session	Ranking test	[96]
Dry-cured hams	Sheep	Appearance, texture, flavor and aroma intensity	8	Trained	Structured scale with 9 points. Quantitative-descriptive analysis	[85]
Dry-cured legs	Sheep and goat	Appearance, taste, flavor and texture	9	Previous experience and specific training	Continuous 10-cm scale. Left anchor-lowest intensity to right anchor-highest intensity	[93]
Cured legs	Sheep and goat	Appearance, aroma, taste and texture	10	Previous experience and specific training	Continuous 10-cm scale. Left anchor-lowest intensity to right anchor-highest intensity	[10]
Sausages	Sheep	Smell, taste, and appearance (control vs. N1, N2 and N3 levels of oregano extract)	14	Trained	Verbal scale with 0 to 6 numerical conversion scale using no, very slight, slight/moderate, moderate, moderate/large, large and very large differences, respectively	[73]
Frankfurters	Goat	Color, odor, taste, hardness and juiciness	7	Trained	Intensity scale test of 9 points for color, odor, taste, hardness and juiciness	[13]
Cooked ham and Deli type sausages	Sheep and goat	Color, taste, odor and firmness	10	Trained	Triangular test and a structured lineal scale (10 cm) for each parameter to evaluate the found difference	[92]
Sheep and goat pâtés	Sheep and goat	Appearance, aroma intensity, taste intensity and texture	8	Trained	Structured but unnumbered 10-cm scale Minimum (no sensation) to maximum (extremely intense sensation)	[94]

**Table 3 foods-09-00960-t003:** Studies on the sensory evaluations of new products from sheep and goat meats using hedonic evaluations with different type of panelists.

Product	Species	Panelists (*n*)	Type of Panelist	Attributes	Scale	Reference
Fermented sausage with different levels of fat addition	Goat	30	Nontrained individuals	Texture, appearance, taste, aroma, overall acceptability	Hedonic nine-points scale. From dislike extremely (1) to like extremely (9)	[107]
Mortadella prepared with different levels of fat	Goat	80	Potential consumers	Appearance, color, odor, flavor, overall acceptability	Hedonic nine-points scale. From dislike extremely (1) to like extremely (9)	[38]
Cooked ham-type pâté	Sheep	50	Nontrained panelists	Overall acceptability	Hedonic nine-points scale. From less preferred than the reference (1) to preferred more than the reference (9)	[58]
Fermented sausage	Sheep	60	Consumers	Liking of flavor	9-point category scale. From dislike extremely (1) to like extremely (9)	[106]
Fresh sausages with and without paprika	Sheep and goat	82	Consumers	Taste, texture, spiciness, overall acceptability	Unstructured 10-cm scale. Anchors at the extremities with do not like (0) to like very much (10)	[92]
Fresh sausages with different levels of pork fat	Sheep and goat	26	Nontrained individuals in 2 sessions	Taste, spicy taste, texture, overall acceptability	Unstructured 10-cm scale. Anchors at the extremities with do not like (0) to like very much (10)	[9]
Patties	Lamb	126	Consumers	Flavor score/liking	Structured hedonic nine-points scale. From ranging from extremely disliking (1) to extremely liking (9)	[96]
“Cecina” dry-cured legs with different fatness levels	Sheep	320	Individuals (Chinese, sub-Saharan, Andeans, Spanish)	Overall acceptability	Hedonic nine-points scale. From dislike extremely (1) to like extremely (9)	[108]
Dry-cured ham	Sheep	375	Individuals	Maturation time, smoking, sodium reduction	No scale	[109]
Coppa	Sheep	375	Individuals	Natural antioxidant, smoking, sodium reduction	No scale	[109]
Frankfurters (25–100%) vs. beef	Goat	60	Consumers (40% male, 60% female)	Appearance, color, odor, taste, texture	CATA (Check-all-that-apply) and rate (0–10) overall acceptance (liking)	[13]
Pâtés	Sheep and goat	25	Individuals	Global acceptability	10-cm scale. Anchors at the extremities with do not like (0) to like very much (10)	[94]

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
