# Peer review of "Sheep and Goat Meat Processed Products Quality: A Review"

_foods, 2020, doi:10.3390/foods9070960_

Round 1
Reviewer 1 Report
Reviewer Evaluation:
This review paper by Teixeira et al assesses caprine and ovine processed meat products with reference to their quality, safety, and organoleptic properties. It is an inciteful and well written review that really considers a vast variety of products globally. However, it lacks critical review and in-depth discussion on the field. Less listing of studies and more critical appraisal is required.
Major Points:
- Line 244 – is there much evidence of bacteriocins being effective for the improvement of food safety? If so what ones are most effective? Nisin? It is important to mention this within the microbiological section.
- The main issue the review seems to have is that it lacks a critical edge. There is a lot of sections that could use a critical eye and discussion. It feels like a lot of studies are being listed without greater critique of the research field. For instance, the reliability of the sensory analysis. Can you provide guidance on how to improve current practices? Are the microbiological standards sufficient currently, is there a particular research avenue that should be followed in that field etc.
- Before the conclusions, a short section on future directions and predictions on the potential growth of the goat and sheep meat industry would be useful to readers.
- Line 115 - pastrami is Romanian origin though?
- Line 129 – I would advise either expand on this point or drop this line. It doesn’t add anything.
- Line 135 – also known as sujuk
- Table 1 – the column on effects studied could be clearer for some. For instance “vacuuming and pressuring on”, on what? Tenderising, but the table doesn’t specify.
- Table 2 and 3 – should the outcomes be reported?
- There is a food safety aspect to this review. You have touched on this topic, but it might be worth expansion on the prevalence of nitrate and nitrite preservation in sheep and goat meat and comment on its safety
Minor Points:
The abstract has some grammatical errors that need attention as highlighted in the minor points below. Respectfully, there is also minor grammatical or sentences structure issues in other parts of the manuscript that the authors should consider addressing. I have tried to help by highlighting a small few. However, they are minor and do not take away from the excellent research presented and of course they are optional.
Line 11 – I would suggest the following edit: Even though sheep and goat processed meat products are not as popular as (name the other meats) and are generally considered not as important. However, they have a very important role in meat consumption around the world.
Line 15 – the use of the word made in both cases is incorrect on this line
Line 16 – processed meat
Line 22 – as a highly acceptable consumer commodity (delete with)
Line 39 – Scandinavian
Line 64 – In modern times, the processing…
Line 111 – replace concretely with ‘mainly’ or ‘concentrated in the ‘
Line 111 – there are different fermented products produced, some of them…
Line 118 – Over the last few years, several…
Line 244 – aw – its better to state moisture. And the w should be subscript
Line 242 – 246 – sentence structure issues
Line 253 - LAB name should be in italics
Line 261-262 – phrasing issue
Line 273 – species name should be in italics
Line 289 – replace was made … with was conducted
Line 293 - leads
Concluding Remarks:
Overall, this review is very insightful and certainly contributes to the field. However, it requires more critical insights and discussion within the article. I look forward to the authors corrections.
Author Response
Responses to Reviewer 1
The authors are grateful for the reviewer's comments and suggestions that enriched the final text.
Reviewer Evaluation:
This review paper by Teixeira et al assesses caprine and ovine processed meat products with reference to their quality, safety, and organoleptic properties. It is an inciteful and well written review that really considers a vast variety of products globally. However, it lacks critical review and in-depth discussion on the field. Less listing of studies and more critical appraisal is required.
Response: The limitation to 5500 words to submit the manuscript limited our text. However we have introduced new sentences with our critical review in all sections.
Major Points:
- Line 244 – is there much evidence of bacteriocins being effective for the improvement of food safety? If so what ones are most effective? Nisin? It is important to mention this within the microbiological section.
Response: according to Ross et al (2002) bacteriocins are known since 1928: “The first observations that led to the discovery of bacteriocins were made by Rogers and Whittier, in England when they discovered that certain lactococcal strains had an inhibitory effect on the growth of other LAB”. They produced a very good revision about fermentation, including bacteriocins. There are several studies, some very recent references about bacteriocins, namely Nisin, but not related with sheep and goat meat products. That is why we did not included a reference to them.
- The main issue the review seems to have is that it lacks a critical edge. There is a lot of sections that could use a critical eye and discussion. It feels like a lot of studies are being listed without greater critique of the research field. For instance, the reliability of the sensory analysis. Can you provide guidance on how to improve current practices? Are the microbiological standards sufficient currently, is there a particular research avenue that should be followed in that field etc.
Response: Text was rewritten to consider the comments
- Before the conclusions, a short section on future directions and predictions on the potential growth of the goat and sheep meat industry would be useful to readers.
Response: Some future trends were pointed out
- Line 115 - pastrami is Romanian origin though?
Response: It was corrected
- Line 129 – I would advise either expand on this point or drop this line. It doesn’t add anything.
Response: The sentence was rewritten
- Line 135 – also known as sujuk
Response: It was introduced in the text.
- Table 1 – the column on effects studied could be clearer for some. For instance “vacuuming and pressuring on”, on what? Tenderising, but the table doesn’t specify.
Response: The table was reformulated according your suggestion
- Table 2 and 3 – should the outcomes be reported?
Response: some outcomes are reported in the text, with the words limit it is not possible to report al the outcomes
- There is a food safety aspect to this review. You have touched on this topic, but it might be worth expansion on the prevalence of nitrate and nitrite preservation in sheep and goat meat and comment on its safety
Response: this was considered in the text
Minor Points:
The abstract has some grammatical errors that need attention as highlighted in the minor points below. Respectfully, there is also minor grammatical or sentences structure issues in other parts of the manuscript that the authors should consider addressing. I have tried to help by highlighting a small few. However, they are minor and do not take away from the excellent research presented and of course they are optional.
Line 11 – I would suggest the following edit: Even though sheep and goat processed meat products are not as popular as (name the other meats) and are generally considered not as important. However, they have a very important role in meat consumption around the world.
Response: The sentence was rewritten according review suggestion
Line 15 – the use of the word made in both cases is incorrect on this line
Response: Changed
Line 16 – processed meat
Response: Changed
Line 22 – as a highly acceptable consumer commodity (delete with)
Response: Deleted
Line 39 – Scandinavian
Response: Changed
Line 64 – In modern times, the processing…
Response: Changed according the suggestion of Reviewer 2
Line 111 – replace concretely with ‘mainly’ or ‘concentrated in the
Response: Changed
Line 111 – there are different fermented products produced, some of them…
Response: The sentence was rewritten
Line 118 – Over the last few years, several…
Response: Changed
Line 244 – aw – its better to state moisture. And the w should be subscript
Response: “w” was converted to subscript.
Line 242 – 246 – sentence structure issues
Response: sentence was rewritten
Line 253 - LAB name should be in italics
Response: LAB does not need to be italic
Line 261-262 – phrasing issue
Response: sentence was rewritten
Line 273 – species name should be in italics
Response: species names were converted to italic
Line 289 – replace was made … with was conducted
Response: replacement done
Line 293 – leads
Response: done
Concluding Remarks:
Overall, this review is very insightful and certainly contributes to the field. However, it requires more critical insights and discussion within the article. I look forward to the authors corrections.
Response: We have introduced some critical insights

Reviewer 2 Report
The manuscript presents interesting data regarding goat and sheep meat products quality in a different region of the world.
General comments
The manuscript presents interesting data regarding goat and sheep meat products quality in different region of the world.
Major revision
Line 111-113 although you discussed some of information regarding meat products of goat and sheep but I would like you add more information in different Himalayan area including china and Pakistan because china has highest population of goats followed by India, Pakistan and Bangladesh.
Line 149-160 very nice information regarding different studies on goat and sheep meat products I would like you to show author name with numbering e.g (Bratcher et al., 2011) (37) has carried a study which clarify that …… and so on.
In table you should add the effect of slaughter age (old and young goat) with storage so I am suggesting a study you should put (Effect of Slaughter Age on Muscle Fiber Composition, Intramuscular Connective Tissue and Tenderness of Goat Meat during Post-Mortem Time) (DOI: 10.3390/foods8110571)
Line 234-284 being a meat scientist I am very impressed from your detailed writing on microbiological quality and safety, as we all know the knowledge regarding goat and lamb meat products and raw meat is very limited. So I am suggesting one point if you could please add some information regarding scrapie, salmonella and botulism, which are considered very common safety aspects of goat and sheep meat. My concern is basic diseases associated with goat and lamb meat consumption eithers its products or raw meat.
Line 304 table 2 is very informative but as dealing with meat mostly consumers concern is about goaty smells in goat meat, I would like you to find some studies in which author had used some ingredients to reduce the goaty smell and you must include it your table.
As you make an great effort to write this manuscript which is clear from the content of manuscript but I would like it to be more perfect and to provide a gate way for coming researcher on goat and sheep meat researcher, so I am suggesting you to add brief description of future innovations, opportunities and how can it industrialized and research opportunities on goat and sheep meat.
Line 2-3 Sheep and goat meat processed products quality. A review. → Sheep and goat meat processed products quality: A review
Line 30 from 1994 to 2004 →from 1994 to 2004 respectively
Line 56-57 the objective of the study is not clear, knowledge is not proper word, the objective should be self-explanatory to the content of the manuscript
Line 59-61 a reference should be given
Line 63 the word modernly should be replaced with currently
Line 66 produced→ products
Line 72- 83 good information if possible provide a reference
Line 86-94 at least provide one reference
Line 100 in Iceland → furthermore in Iceland
Line 211 please check the table 1, column 20 and justify your information (Beef, sheep and goat)
Line 232 PDO and PGI brands→ abbreviations should be given”
Line 245 please check English and comma and full stop (by ripening or salting) [75; 76]; 2) Food safety improvement through pathogens inhibition.
Line 322 To soothe→ reduce the space
Line 329 please justify your sentence
Line 344 sheepmeat → sheep meat
Line 401Processed products as sausages → Processed products such as sausages
451-456 In reference list reference no 11 and 12 should be justify
Line 533-535 reference 44 should be justify
Line 550-552 reference no 51 should be justify
Line 691-194 reference no 102 should be justify
Line 703-705 reference no 106 should be justify
Author Response
Responses to Reviewer 2
The authors are grateful for the reviewer's comments and suggestions that enriched the final text
The manuscript presents interesting data regarding goat and sheep meat products quality in a different region of the world.
General comments
The manuscript presents interesting data regarding goat and sheep meat products quality in different region of the world.
Major revision
Line 111-113 although you discussed some of information regarding meat products of goat and sheep but I would like you add more information in different Himalayan area including china and Pakistan because china has highest population of goats followed by India, Pakistan and Bangladesh.
Response: Although China, India and Pakistan are very important sheep and goat producers there are a scarcity of bibliography, particularly scientific papers about research on meat processed products. However, we have introduced some more information about those countries
Line 149-160 very nice information regarding different studies on goat and sheep meat products I would like you to show author name with numbering e.g (Bratcher et al., 2011) (37) has carried a study which clarify that …… and so on.
Response: We have found no particular reason why identifying this citation differently from all others
In table you should add the effect of slaughter age (old and young goat) with storage so I am suggesting a study you should put (Effect of Slaughter Age on Muscle Fiber Composition, Intramuscular Connective Tissue and Tenderness of Goat Meat during Post-Mortem Time) (DOI: 10.3390/foods8110571)
Response: As the suggested study does not follow the scope iof our revision (met processed products) we did not find a particular reason to accept the reviewer's suggestion
Line 234-284 being a meat scientist I am very impressed from your detailed writing on microbiological quality and safety, as we all know the knowledge regarding goat and lamb meat products and raw meat is very limited. So I am suggesting one point if you could please add some information regarding scrapie, salmonella and botulism, which are considered very common safety aspects of goat and sheep meat. My concern is basic diseases associated with goat and lamb meat consumption eithers its products or raw meat.
Response: thank you for you kind comment, but as you may understand this review is related to sheep and goat meat products and not raw or fresh meat, and as far as it was possible to find in the bibliography consulted no issues regarding the safety aspects you refer were found since conservatives were used to prevent them. Maybe in future papers that can be considered
Line 304 table 2 is very informative but as dealing with meat mostly consumers concern is about goaty smells in goat meat, I would like you to find some studies in which author had used some ingredients to reduce the goaty smell and you must include it your table.
Response: it is referred in the text and in the tables the use of additives to mask the goat peculiar odor and not only odor, but flavor. For example, the authors (Lu, Y.; Young, O.A.; Brooks, J.D. Physicochemical and sensory characteristics of fermented sheep meat sausage. Food Sci. Nutr. 2014, 2, 669–675. DOI: 10.1002/fsn3.151) refer that the goal was achieved. However, sometimes is not the odor that people don’t like but the prejudice they have and think they don’t like. Sometimes when you make a blind proof of sheep and goat meat people like the sample, and when you tell them what it is, they get surprised. Some people like the peculiar odor of sheep ang goat meat and if you mask it, the product lose their identity.
As you make an great effort to write this manuscript which is clear from the content of manuscript but I would like it to be more perfect and to provide a gate way for coming researcher on goat and sheep meat researcher, so I am suggesting you to add brief description of future innovations, opportunities and how can it industrialized and research opportunities on goat and sheep meat.
Response:
Line 2-3 Sheep and goat meat processed products quality. A review. → Sheep and goat meat processed products quality: A review
Response: The title was modified according the suggestion
Line 30 from 1994 to 2004 →from 1994 to 2004 respectively
Response: The sentence was rewritten according the suggestion
Line 56-57 the objective of the study is not clear, knowledge is not proper word, the objective should be self-explanatory to the content of the manuscript
Response: The sentence was rewritten according the reviewer suggestion
Line 59-61 a reference should be given
Response: Authors think that is from the common knowledge and did not need a reference
Line 63 the word modernly should be replaced with currently
Response: Changed according the suggestion
Line 66 produced→ products
Response: Changed
Line 72- 83 good information if possible provide a reference
Response: This is known to one of the authors who is the coordinator of the referred Red CYTED.
Line 86-94 at least provide one reference
Response: This is known to one of the authors who is the coordinator of the referred Red CYTED.
Line 100 in Iceland → furthermore in Iceland
Response: Changed according the reviewer suggestion
Line 211 please check the table 1, column 20 and justify your information (Beef, sheep and goat)
Response: Checked
Line 232 PDO and PGI brands→ abbreviations should be given”
Response: The names of abbreviations were introduced when appears for the first time (lines 59-60 in the corrected version)
Line 245 please check English and comma and full stop (by ripening or salting) [75; 76]; 2) Food safety improvement through pathogens inhibition.
Response: checked
Line 322 To soothe→ reduce the space
Response: checked
Line 329 please justify your sentence
Response: checked
Line 344 sheepmeat → sheep meat
Response: both are possible
Line 401Processed products as sausages → Processed products such as sausages
Response: done
451-456 In reference list reference no 11 and 12 should be justify
Response: checked
Line 533-535 reference 44 should be justify
Response: checked
Line 550-552 reference no 51 should be justify
Response: checked
Line 691-194 reference no 102 should be justify
Response: checked
Line 703-705 reference no 106 should be justify
Response: checked

Round 2
Reviewer 1 Report
The authors have answered my queries. I think they have given a great overview of sheep and goat products.
Good luck with your future studies.
Reviewer 2 Report
The author made changes accordingly, so the paper is being accepted